# How Vulnerable Are Patients with COPD to Weather Extremities?—A Pilot Study from Hungary

**DOI:** 10.3390/healthcare10112309

**Published:** 2022-11-18

**Authors:** Gergely Márovics, Éva Pozsgai, Balázs Németh, Szabolcs Czigány, Szilvia Németh-Simon, János Girán

**Affiliations:** 1Department of Public Health Medicine, University of Pécs Medical School, 12. Szigeti St., 7624 Pécs, Hungary; 2Department of Primary Health Care, University of Pécs Medical School, 2. Rákóczi St., 7623 Pécs, Hungary; 3Department of Physical and Environmental Geography, Institute of Geography and Earth Sciences, University of Pécs Faculty of Sciences, 6. Ifjúság St., 7624 Pécs, Hungary; 4Department of Emergency Medicine, University of Pécs Medical School, 13. Ifjúság St., 7624 Pécs, Hungary

**Keywords:** weather extremes, meteorological parameters, emergency department visits, COPD

## Abstract

Chronic obstructive pulmonary disease (COPD) is one of the most common causes of death globally, with increasing prevalence and years lived with disability (YLD). We aimed to investigate how extreme weather conditions were associated with the number of daily COPD-related emergency visits. We collected data regarding the number of daily emergency department (ED) visits made by patients with COPD in 2017, along with all relevant daily meteorological data for the same year. An analysis of the relationship between the number of COPD-related ED visits and extreme meteorological events was carried out. Extremely low temperatures (OR = 1.767) and dew points (OR = 1.795), extremely high atmospheric pressure (OR = 1.626), a high amount of precipitation (OR = 1.270), and light wind speed (OR = 1.560) were identified as possible risk factors for a higher number of COPD-related ED visits. In contrast, extremely high temperatures (OR = 0.572) and dew points (OR = 0.606) were found to be possible protective factors for COPD-related ED visits. By determining the meteorological risk factors for a high number of COPD-related ED visits, our study may help provide invaluable data for identifying vulnerable patient groups based on weather events, thus making more optimal capacity planning at the ED possible.

## 1. Introduction

Climate and weather have a significant impact on the biological functions and health of the human body. As we are facing the wide-ranging impacts of global climate change, the relevance of its effects on humans is increasing. Since climate change affects meteorological parameters and weather patterns, they may, in turn, influence chronic disease-related morbidity and mortality rates [1,2,3,4]. Studies have suggested that extreme high temperatures and heatwaves elevate the risk of mortality in patients with cardiovascular, cerebrovascular, and respiratory diseases, as well as in the elderly population [5]. According to a literature review, increased demand for medical care for people with renal diseases and heat-related illnesses has been observed in various geographical areas [6].

In addition to the heat, cold spells have also been shown to influence the frequency of certain non-communicable diseases. Studies have shown that extremely low temperatures in certain geographical regions can increase the number of admissions in an emergency department [7]. A high diurnal temperature range was also reported to be a risk factor for increased cardiovascular and respiratory-related hospital admissions [8,9]. It has also been suggested that extremely low temperatures significantly increase the risk of respiratory symptom-related emergency room visits [10]. The association between weather parameters and hospital admissions has been proven, and this connection seems to be stronger in the case of respiratory diseases [11]. Furthermore, instead of focusing solely on temperature-related factors, studying other weather-related extremes—that can also affect health—would be more beneficial.

Chronic obstructive pulmonary disease (COPD) causes more than 3 million deaths a year, comprises 6% of all deaths, and is the third most common cause of death globally [12]. In 2015, the global prevalence of COPD was almost 175 million, an increase of 17% compared to 2005. During these ten years, the age-standardized rates (ASR) for prevalence showed a decrease of 5.8%. Looking at the same period, the years lived with disability (YLD) increased by 16.2% (12,047 years in 2015), while the ASR for YDL decreased by 5.9% [13]. Since millions of people smoke worldwide, they constitute a large risk group for COPD. Other risk factors include inhalation of various types of smoke (including second-hand smoking) and ambient particulate matter [14].

The hospital admission of patients with COPD was found to be more frequent during the wintertime according to a previous analysis [15]. The same conclusions were drawn from a population-based study, where the highest number (37.2%) of hospital admissions due to COPD was observed during the winter months, followed by autumn, spring, and summer [16]. A German study reported that patients hospitalized due to the exacerbation of COPD during the winter were significantly older than patients hospitalized in the summer (*p* = 0.04) [17]. Extreme cold and hot temperatures were also associated with increased morbidity in COPD [18]. Evidence showed that the cumulative relative risks of COPD admissions were 1.06 during the hot season (30 °C vs. 25 °C) and 1.64 during the cold season (12 °C vs. 21 °C) [19]. Furthermore, according to a study in Taiwan, a 1 °C decrease in temperature increased the exacerbation rate of COPD by 0.8% [20]. While another report in Taichung City, Taiwan, showed that there was a negative association between the average daily temperature and emergency admissions for COPD [9].

Investigating the effects of extreme meteorological parameters on the frequency of hospital visits is important because it can provide valuable information that can serve as the basis for designing preventive measures and capacity planning for healthcare providers.

Therefore, the goal of our study was to determine how the extremes of certain meteorological parameters (temperature, dew point, atmospheric pressure, wind speed, and precipitation) affected the number of visits made by patients with COPD to the local emergency department (ED).

## 2. Materials and Methods

### 2.1. Setting

The study was conducted in Pécs, which is a university city located in the south-western part of Hungary with approximately 130,000 inhabitants. The city’s microclimate is sunnier and warmer than the national average, but the annual precipitation is similar to the Hungarian national average. A meteorological station, which is part of the monitoring network of the Hungarian Meteorological Service, has been operating at the airport near the city since 1956. The ED is part of a regional medical center, the University of Pécs Clinical Center, and its average annual patient turnover is around 50–60,000 patients.

### 2.2. Study Design

Our study received ethical approval from the Regional Ethical Committee (Reference Number: 8287-PTE2020).

Meteorological data for the study were obtained from two separate sources. Meteorological data recorded by a certified monitoring station at Pécs-Pogány airport, on the outskirts of the city, were obtained from the National Oceanic and Atmospheric Administration website. The coordinates of the station are 45.991° N 18.241° E, 203 m above the Baltic Sea (ABS). Data from another meteorological station located at the Ifjúság Street Campus of the University of Pécs, 46.078° N, 18.207° E, 174 m ABS, was also used. It is a validated monitoring station operated by the Department of Physical and Environmental Geography, Institute of Geography and Earth Sciences, University of Pécs Faculty of Sciences.

The weather data of the city of Pécs and the data of patients presenting to the ED between 1 January 2017–31 December 2017 were gathered. The data from the two meteorological stations in Pécs were used to eliminate the differences between the values in the center and on the outskirts of the city. Data collection regarding meteorological parameters included the daily minimum, maximum, and average temperature values in degrees Celsius (°C); the precipitation in millimeters (mm); dew point in degrees Celsius (°C); the atmospheric pressure data at the station level in hectopascal (hPa); and the average wind speeds in meters per second (m/s) for each day of the studied year. Data collection regarding patient visits was carried out using the University of Pécs Clinical Center’s e-MedSolution database. The following patient-related data were gathered for patients 18 years of age or older: sex; date of birth; date of admission to the ED and discharge; referent ID; diagnosis following ED admission. The 10th revision of the International Classification of Diseases (ICD-10) codes were used to identify the diagnoses.

### 2.3. Data Analysis

Regarding weather parameters, data were dichotomized: all variables were split by the 5th, 10th, 90th, and 95th percentile values to indicate extreme conditions. In the cases of the 5th and 10th percentiles, the values that were lower than or equal to the cut-off value were coded with value 1, the others with value 2. In the cases of the 90th and 95th percentile, the values, which were greater than or equal to the cut-off value were coded with value 1, the others coded with value 2. That indicated four different transformed dichotomic variables for each meteorological parameter. The exact cut-off values are shown in Table 1.

Regarding the data from the ED visits, the data series was split into two series according to the daily averages: days with at least 141 cases registered in the ED were considered as “high number of daily ED visits” and coded with value 1, and other days were considered as “low number of daily ED visits” and coded with value 2 in the database. When any of the six ICD-10 based diagnosis codes available for COPD (J4400, J4410, J4480, or J4490) were documented for the patients: the patients were coded with value 1, and when the patients did not have any of these four ICD-10 codes, they were coded with value 2. Regarding the data of the COPD cases, the data series was split into two series according to the daily averages: days with at least 3 COPD cases registered in the ED were to have considered a “high number of daily COPD-related ED” and coded with value 1 in the database, while a daily COPD-related ED visit number below 3 was considered to have a “low number of daily COPD-related ED visits” for the purpose of analysis.

We intended to investigate whether the extreme values of the meteorological parameters affected the overall number of emergency visits and the health status of patients suffering from COPD. Therefore, the weather parameters were considered independent variables, while the number of ED visits and COPD-related ED visits were the dependent variables.

### 2.4. Statistical Analysis

The data were subjected to statistical analysis using SPSS 28 software. The data analysis included the use of frequency tables, contingency tables, and correlation matrixes. Moreover, the chi-square test and risk estimation were performed to examine the stochastic associations and present the strength of association between possible risk factors and outcomes. Differences and associations were considered statistically significant if *p* < 0.05.

## 3. Results

### 3.1. Local Climatological Data of the Studied City (Pécs) in 2017

The average temperature was 11.73 °C, with daily mean temperatures ranging from −10.83 °C to 29.72 °C. The lowest recorded temperature was −14.39 °C and the highest was 36 °C. The daily mean diurnal temperature range was 9.38 °C; the day with the highest diurnal temperature range was 17.61 °C, while the lowest was 1 °C. The average measured dew point was 5.36 °C and varied between −20.28 °C and 19.4 °C during the year. The total rainfall was 646.684 mm, with 59.69 mm recorded on the wettest day. Almost three-quarters of the year (73.4%) was dry: there were 268 days without rainfall. The mean precipitation was 1.72 mm. The average atmospheric pressure measured at the station level was 993.93 hPa and varied between 970.5 hPa and 1012.4 hPa. The mean value of the average daily wind speed was 11.03 m/s and varied between 2.04 m/s and 32.04 m/s (Table 2).

### 3.2. Demographic and Clinical Characteristics of Patients Visiting the ED in the Studied City (Pécs) in 2017

Altogether 51,436 ED visits were registered in 2017. The descriptive statistics of the daily emergency visits and the COPD cases are illustrated in Table 3. The mean age of the patients was 58.72 years, and the median was 62 years. A total of 48.8% of the patients were male and 51.2% were female. The mean age of patients with COPD was 67.71 years, with a median age of 68 years. In contrast, the mean age of patients without COPD was lower, 58.52 years, with a median of 61 years. From the total number of COPD-related emergency visits, more than half, 50.7%, were patients aged between 60–74 years. 36.9% of the COPD patients were male, and 63.1% were female. Females between 65–74 years old constituted most of the COPD cases, representing 33% of the total number of COPD-related emergency visits.

### 3.3. Association between Extremes of Meteorological Parameters and the Number of ED Visits

We analyzed the relationship between the extremes of individual meteorological parameters and the number of ED visits made by patients. We identified some low temperature-related events as an increased risk for a higher number of ED visits per day. The results of the analysis can be found in the Appendix A.

### 3.4. Association between Extremes of Meteorological Parameters and Number of COPD-Related ED Visits

We analyzed the relationship between the extreme values of individual meteorological parameters and the number of ED visits made by patients with COPD. The odds of a high number of COPD-related ED visits per day were more than 75% higher when the average temperature was at or below the 5th percentile and when the dew point was at or below the 5th percentile (OR_AT_^5^ = 1.767 [CI95% _AT_^5^ 1.428–2.187] and OR_DP_^5^ = 1.795 [CI95% _DP_^5^ 1.445–2.228], respectively. The odds ratio of a high number of daily COPD-related ED visits was 1.743 (CI95% 1.405–2.161), when the maximum temperature was at or below the 5th percentile, and 1.698 (CI95% 1.360–2.119) when the minimum temperature was at or below the 5th percentile. The odds of a high number of daily COPD-related ED visits were approximately 60% higher when the average temperature was at or below the 10th percentile, when the minimum temperature was at or below the 10th percentile, when the station level pressure was at or above the 95th percentile, and when the dew point was at or below the 10th percentile (OR_AT_^10^ = 1.604 [CI95%_AT_^10^ 1.353–1.900], OR_MNT_^10^ = 1.611 [CI95%_MNT_^10^ 1.364–1.903], OR_STP_^95^ = 1.626 [CI95%_STP_^95^ 1.309–2.021], OR_DP_^10^ = 1.646 [CI95% _DP_^10^ 1.392–1.946]), respectively. Regarding atmospheric pressure and wind speed, the odds ratios indicated an approximately 50% higher risk. The odds ratio was 1.513 (CI95% 1.275–1.795) for a high number of daily COPD-related ED visits when the station-level pressure was at or above the 90th percentile, and 1.560 (CI95% 1.248–1.948), when the measured average wind speed was at or below the 5th percentile. We found that the odds of a high number of daily COPD-related ED visits did not reach 50% when the maximum temperature was at or below the 10th percentile (OR_MXT_^10^ = 1.450 [CI95% _MXT_^10^ 1.208–1.740]). Our analysis also showed a 27% higher risk (OR_PRCP_^90^ = 1.270 [CI95% _PRCP_^90^ 1.060–1.522]) for a high number of daily COPD-related ED visits when the amount of precipitation was at or above the 90th percentile. Therefore, these results suggest that the mentioned meteorological parameters were risk factors for a high number of daily COPD-related ED visits (Figure 1).

The phi correlation coefficient showed a statistically significant weak relationship between the high number of daily COPD-related ED visits and some of the weather parameters shown in Table 3. The coefficients varied between 0.026 (when the dew point was at or below the 10th percentile) and 0.011 (when precipitation was at or above the 90th percentile) (Table 4). We did not find a statistically significant relationship between the high number of daily COPD-related ED visits and the rest of the studied weather parameters.

Besides these risk factors, we could identify some protective factors as well. The weather parameter that had the lowest odds ratio for increased number of COPD-related ED visits was the minimum temperature at or above the 95th percentile. This parameter lowered the risk by approximately 40% (OR_MNT_^95^ = 0.572 [CI95% _MNT_^95^ 0.406–0.807]). Similar odds ratios were found when the dew point was at or above the 90th percentile (OR_DP_^90^ = 0.606 [CI95%_DP_^90^ 0.477–0.771]) and when the average temperature was at or above the 95th percentile (OR_AT_^95^ = 0.611 [CI95% _AT_^95^ 0.433–0.862]). When the minimum temperature was at or above the 90th percentile, we detected an approximately 35% lower risk for a high number of COPD-related ED visits (OR_MNT_^90^ = 0.639 [CI95%_MNT_^90^ 0.503–0.812]). Similar odds ratios were calculated when the maximum temperature and the dew point were at or above the 95th percentile (OR_MXT_^95^ = 0.659 [CI95% _MXT_^95^ 0.474–0.917] and OR_DP_^95^ = 0.661 [CI95% _DP_^95^ 0.479–0.912]). The odds ratios of the average and maximum temperatures at or below the 90th percentile showed more than 30% lower risks for a high number of COPD-related ED visits (OR_AT_^90^ = 0.677 [CI95%_AT_^90^ 0.536–0.857] and OR_MXT_^90^ = 0.681 [CI95% _MXT_^90^ 0.539–0.860]) (Figure 2).

The phi correlation coefficient showed a statistically significant weak inverse relationship between the high number of daily COPD-related ED visits and some of the weather parameters shown in Table 5. The coefficients varied between −0.140 (when the average temperature was at or above the 90th percentile) and −0.011 (when the maximum temperature and dew point were at or above the 95th percentile) (Table 5). We did not find a statistically significant relationship between the high number of daily COPD-related ED visits and the rest of the studied weather parameters.

## 4. Discussion

To our knowledge, this is the first study to investigate the effect of the microclimate setting—including all available meteorological parameters and weather extremities—on the number of daily ED visits made by patients with COPD.

Our study showed that extremes of low temperature and dew point and high atmospheric pressure were risk factors for an increased number of daily emergency visits. The range of odds ratios was between 2.690 and 4.929 in the case of cold-temperature-related emergency visits. This, therefore, indicated that in extreme cold weather conditions, the risk of a higher-than-average number of daily emergency visits can increase almost fivefold. Our results are in line with other studies investigating the association between cold temperature events and the frequency of emergency visits [21,22,23]. Similar conclusions were drawn in a systematic review, which reported that the fraction of respiratory morbidity attributable to cold was 12.2% in high-income countries and 25.7% in a middle-income country [24].

Our analysis demonstrated that extreme low temperatures and dew points, light wind speeds, high precipitation, and extreme high atmospheric pressure were risk factors for a higher number of COPD-related emergency visits. In agreement with our results, a literature review clearly showed that lower temperatures had a significant negative effect on patients with COPD [25]. A study from Poland reported significant negative correlations between the average temperature (r = −0.577), minimum temperature (r = −0.526), and dew point (r = −0.589) and an increased number of COPD-related ED visits with a significance level of *p* < 0.01 [26]. In line with our results, a study in London also showed that a higher number of COPD-related emergency visits could be expected when there was high rainfall or the wind speed was light [27]. Extreme atmospheric circumstances were also associated with an elevated risk for COPD-related emergency visits [28]. We found that although the odds ratios were typically lower compared to the daily emergency visits (1.270 vs. 1.795), there was still an increased risk in the case of extreme atmospheric pressure. Regarding certain temperature-related parameters, our results showed almost twice the odds for a higher number of COPD-related ED visits. Similar to our study, the relative risk of extremely low daily mean temperatures in the case of COPD admissions was 1.55 in a Chinese study [29]. Our results emphasize the significantly higher morbidity burden of COPD patients and patients in general on ED staff in extreme meteorological conditions.

On the other hand, high temperatures and dew points were shown to be protective factors for COPD-related ED visits. Thus, interestingly, certain weather parameters, such as extremes of high minimum temperatures, can decrease the risk of a high number of COPD-related ED visits by more than 40%.

Our study also revealed that the majority of the COPD-related ED visits belonged to patients older than the average age of the patients who visited the ED in general, and female dominance could be observed among the patients with COPD. The same observation was made in China, where a study showed that the female population with COPD could be more vulnerable to cold temperatures [29]. In Taiwan, a study reported a higher risk for COPD-related ED visits among patients 65 years old or above with a long cold effect (>14 days) than for patients below 65 years with a shorter cold effect (lag 0–14 days) [20]. In line with these results, a study in Hong Kong showed that the effect of low temperatures lasted longer and that the older population was more sensitive to it [19]. In a recent Chinese study, low ambient temperature was a risk factor for acute exacerbation of COPD among older patients [30]. On the other hand, an investigation in Guangzhou, China, found inconsistent results and demonstrated higher relative risks for 1°C of temperature increase for males and people aged 0–64 years [31]. This is of particular interest since some of the previous studies had found conflicting results about the increased risk of older female COPD patients’ ED visits compared to other patient groups. Thus, by outlining a group of patients more prone to ED visits under certain meteorological conditions, our observations indicate that they may be a special risk group.

The outcomes of our study draw attention to how the emergency department’s capacity planning could possibly be supported by weather forecasts. By being aware of meteorological variations that may pose increased risks for patients with certain diseases, such as COPD, ED capacity planners could make informed decisions about the distribution of ED staff, which, in turn, might improve the quality of care for these patients. Further studies are needed to specify vulnerable subpopulations during extreme weather conditions to possibly help the work of ED capacity planners and staff. Furthermore, family doctors and/or doctors treating COPD patients could utilize meteorological information by educating their patients regarding the possible effects of certain meteorological parameters on the symptoms of their disease. Moreover, developing a web-based application to warn patients with COPD and give them guidance due to the harmful weather events could be another possible way to transfer information.

Our investigation has several limitations. Firstly, since it was a pilot study, we only investigated one year of data. Secondly, we investigated data relating to morbidity rates, which is always a more complicated task than investigating mortality alone because outcomes are not always clear-cut and several confounding factors can affect the results, such as inaccurate data recording or the presence of comorbidities. Thirdly, due to the nature of the recorded data, it is not always possible to unambiguously identify the cause of the ED visit (for e.g., the patient may visit the ED either because of the worsening of their COPD symptoms or due to similar symptoms of another comorbidity like ischemic heart disease or high blood pressure). Finally, our analysis would have been more precise if the emergency visit reports could have been tracked hourly throughout the day instead of being included in one daily summarized report. This option would have given an exact picture of how the changes in the meteorological parameters during the day affected the ED visits. Acute diseases—for example, accidents because of bad weather conditions—and chronic diseases can also increase the daily case numbers due to seasonality.

## 5. Conclusions

Using a novel methodological approach, our study provided data regarding the effect of a wide range of meteorological parameters on patients with COPD. In contrast to previous studies, which only focused on the effect of just one or maybe a few meteorological parameters, our investigation analyzed the effect of several meteorological parameters on the number of ED visits and COPD-related ED visits. Low temperatures, low dew points, and a high atmospheric pressure were found to be risk factors for a higher number of ED visits. Low temperatures and dew points, light wind speeds, a high amount of precipitation, and a high atmospheric pressure were found to be risk factors for a higher number of COPD-related ED visits. Low temperatures, low dew points, and a high atmospheric pressure appeared to be shared risk factors for an increased number of general- and COPD-related ED visits. However, the associations observed regarding COPD-related ED visits were more reliable due to a more detailed analysis of these data. Light wind speeds and a high amount of precipitation were identified as novel risk factors for a higher number of COPD-related ED visits. On the other hand, certain parameters, such as high temperatures and dew points, strong winds, and a low atmospheric pressure, were shown to be protective factors for COPD-related ED visits.

Our study highlighted the importance of weather variations on morbidity and consequent visits to the ED. Further investigations regarding the health effects of various meteorological parameters are warranted to help identify risk groups and develop adequate preventive actions for patients in different healthcare settings. 

## Figures and Tables

**Figure 1 healthcare-10-02309-f001:**
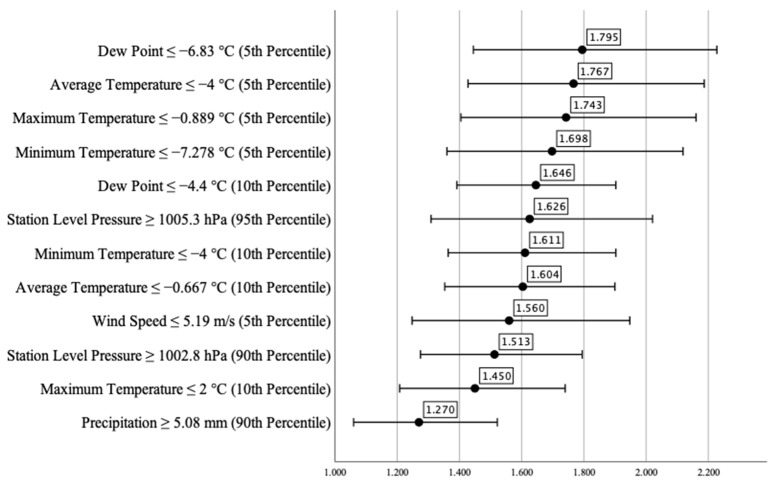
Risk estimation of the effect of extreme meteorological parameters on the number of COPD-related emergency visits per day, *p* < 0.05.

**Figure 2 healthcare-10-02309-f002:**
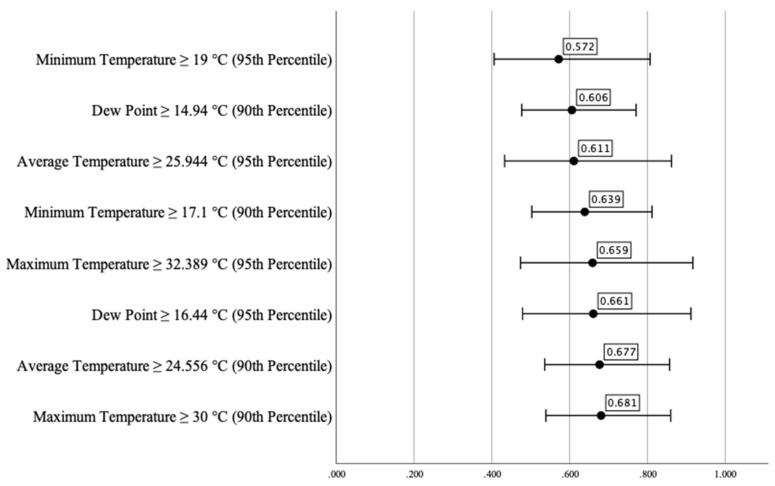
Risk estimation of the effect of extreme meteorological parameters on the number of COPD-related emergency visits per day, *p* < 0.05.

**Table 1 healthcare-10-02309-t001:** The basis for the dichotomization of meteorological parameters. (Abbreviation: Avg.Temp.: average temperature, Max.Temp.: maximum temperature, Min.Temp.: minimum temperature, DTR: diurnal temperature range, Dew.P.: dew point, PRCP: precipitation, STP: station-level air pressure, Avg.WdSp.: average wind speed).

	Avg.Temp. (°C)	Max.Temp. (°C)	Min.Temp. (°C)	DTR (°C)	DewP. (°C)	PRCP (mm)	STP (hPa)	Avg.WdSp. (m/s)
5th percentile	−4	−0.889	−7.278	2.778	−6.833	0	983.3	5.186
10th percentile	−0.667	2	−4	4.111	−4.444	0.1	985	6.297
90th percentile	24.056	30	17.111	14	14.944	5.08	1002.8	17.224
95th percentile	25.944	32.389	19	15	16.444	9.906	1005.3	20.372

**Table 2 healthcare-10-02309-t002:** Local climatological data of Pécs in 2017. (Abbreviation: Avg.Temp.: average temperature, Max.Temp.: maximum temperature, Min.Temp.: minimum temperature, DTR: diurnal temperature range, Dew.P.: dew point, PRCP: precipitation, STP: station-level air pressure, Avg.WdSp.: average wind speed).

	Avg.Temp. (°C)	Max.Temp. (°C)	Min.Temp. (°C)	DTR (°C)	DewP. (°C)	PRCP (mm)	STP (hPa)	Avg.WdSp. (m/s)
Mean	11.73	16.39	7	9.39	5.36	1.71	993.93	11.03
Median	12	17	7.28	9.61	5.33	0	993.6	10
Mode	−0.33	23	4	12.39	1.11	0	993.9	7.22
Standard deviation	9.15	10.33	8.01	3.64	7.17	5.46	6.71	4.75
Min	−10.83	−7.39	−14.39	1	−20.28	0	970.5	2.04
Max	29.72	36	24.39	17.61	19.44	59.69	1012.4	32.04

**Table 3 healthcare-10-02309-t003:** Descriptive statistics of the daily emergency department visits and the COPD cases.

	Daily Emergency Visits	COPD Cases	Relative Frequency of COPD (%)
Sum	51,436	1076	2.1
Mean	140.92	3.32	2.37
Median	137	3	2
Standard deviation	22.7	2.19	1.55
Min	92	0	0
Max	229	12	7.94

**Table 4 healthcare-10-02309-t004:** Relationship between the extreme meteorological parameters and the number of COPD-related emergency visits per day, *p* < 0.05. (Abbreviation: temperature and dew point values in degrees Celsius (°C); the precipitation in millimeters (mm); dew point in degrees Celsius (°C); atmospheric pressure data at the station level in hectopascals (hPa); and the wind speeds in meters per second (m/s).

Weather Parameter	Phi Correlation Coefficient
Dew Point ≤ −4.4 °C (10th Percentile)	0.026
Minimum Temperature ≤ −4 °C (10th Percentile)	0.025
Average Temperature ≤ −0.667 °C (10th Percentile)	0.024
Dew Point ≤ −6.83 °C (5th Percentile)	0.024
Average Temperature ≤ −4 °C (5th Percentile)	0.023
Maximum Temperature ≤ −0.889 °C (5th Percentile)	0.023
Minimum Temperature ≤ −7.278 °C (5th Percentile)	0.021
Station Level Pressure ≥ 1002.8 hPa (90th Percentile)	0.021
Station Level Pressure ≥ 1005.3 hPa (95th Percentile)	0.020
Maximum Temperature ≤ 2 °C (10th Percentile)	0.018
Wind Speed ≤ 5.19 m/s (5th Percentile)	0.017
Precipitation ≥ 5.08 mm (90th Percentile)	0.011

**Table 5 healthcare-10-02309-t005:** Relationship between the extreme meteorological parameters and the number of COPD-related emergency visits per day, *p* < 0.05.

Weather Parameter	Phi Correlation Coefficient
Average Temperature ≥ 24.556 °C (90th Percentile)	−0.140
Maximum Temperature ≥ 30 °C (90th Percentile)	−0.140
Average Temperature ≥ 25.944 °C (95th Percentile)	−0.130
Dew Point ≥ 14.94 °C (90th Percentile)	−0.018
Minimum Temperature ≥ 17.1 °C (90th Percentile)	−0.016
Minimum Temperature ≥ 19 °C (95th Percentile)	−0.014
Maximum Temperature ≥ 32.389 °C (95th Percentile)	−0.011
Dew Point ≥ 16.44 °C (95th Percentile)	−0.011

## Data Availability

Not applicable.

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
