# Peer review of "How Vulnerable Are Patients with COPD to Weather Extremities?—A Pilot Study from Hungary"

_healthcare, 2022, doi:10.3390/healthcare10112309_

Round 1

Reviewer 1 Report

The authors investigated how extreme weather conditions are associated with the number of daily COPD-related emergency visits and identified risk factors (extremely low temperature and low dew point, extremely high atmospheric pressure and high amount of precipitation, and light wind speed) and possible protective factors  (extremely high temperature and high dew point).

I suggest accepting this article for the publication in Healthcare  after minor revisions.

Yet, a rather significant remark (on the border between minor and major point) is that the article lacks some recent references. Significant improvement of the article can be achieved by reformulating the introduction/discussion in such a way as to include the results of recent research. Authors are asked to pay special attention to this remark and correct the article accordingly.

For a clearer presentation of the results, it is suggested to better define the dew point values in lines 20 and 25 (i.e. low dew point and high dew point).

Please insert table legend regarding Table 3 and explain the abbreviations used.

In the line 332 the authors used formulation „important subset of patients“. That statement needs to be reformulated because there are no important and less important groups of patients, all patients must be considered equally important.

In the Conclusions please clearly state which risk factors are shared by COPD and non-COPD patients, and which are unique to COPD patients.

Author Response

Dear Reviewer #1,

I would like to thank you for the assessment of our manuscript entitled “How Vulnerable are Patients with COPD to Weather Extremities? – A Pilot Study from Hungary” (healthcare-2015220) and for your insightful suggestions to improve the manuscript. We have addressed each comment and made the requested modifications, where needed.

Yet, a rather significant remark (on the border between minor and major point) is that the article lacks some recent references. Significant improvement of the article can be achieved by reformulating the introduction/discussion in such a way as to include the results of recent research. Authors are asked to pay special attention to this remark and correct the article accordingly.

Thank you for your comment. In agreement with the Reviewer, therefore we have added some recent references to our manuscript. (Reference No.10 in Line 45, No.11 in Line 46, No.25 in Line 277, No.26 in Line 279, and No.30 in Line 305)

For a clearer presentation of the results, it is suggested to better define the dew point values in lines 20 and 25 (i.e., low dew point and high dew point).

Thank you for your comment. We agree with the Reviewer’s suggestion and have added more defined dew point values. Due to word count limitations, we have placed them in the Methods section instead of the Abstract. (Table 1. in Line 121.)

Please insert table legend regarding Table 3 and explain the abbreviations used.

Thank you for your comment. As the Reviewer requested, we have inserted the table legend for Table 3 (which is Table 4 now) and explained the abbreviations.

In the line 332 the authors used formulation „important subset of patients”. That statement needs to be reformulated because there are no important and less important groups of patients, all patients must be considered equally important.

Thank you for your comment. As the Reviewer suggested, we have corrected the sentence. (Line 345.)

In the Conclusions please clearly state which risk factors are shared by COPD and non-COPD patients, and which are unique to COPD patients.

Thank you for your comment. In agreement with the Reviewer, we have rewritten the Conclusions section. (Line 352.)

Reviewer 2 Report

Title: How Vulnerable are Patients with COPD to Weather Extremities? A Pilot Study from Hungary

Abstract: Abstract must include some results (average, p, etc)

Introduction: good

Methodology: good

Discussion:

Is there any literature from Europe or region?

There are comparing with China and Taiwan, not from the region

(Javorac, J. What Are the Effects of Meteorological Factors on Exacerbations of COPD? Atmosphere 2021,12, 442. https://doi.org/10.3390/atmos12040442

 Conclusion: Concisely  

Literature: very good

2017-2022 – 12/25=48 %

2012-2016 – 9/25 = 36 %

2011≥ - 4/25 = 16 %

The authors cover all aspects of COPD to the Weather Extremities. It would be better to compare to some papers/reviews from the region

Author Response

Dear Reviewer #2,

I would like to thank you for the assessment of our manuscript entitled “How Vulnerable are Patients with COPD to Weather Extremities? – A Pilot Study from Hungary” (healthcare-2015220) and for your insightful suggestions to improve the manuscript. We have addressed each comment and made the requested modifications, where needed.

Abstract: Abstract must include some results (average, p, etc).

Thank you for your comment. In agreement with the Reviewer, we have included some specific results. (Lines 19-23.)

Discussion: Is there any literature from Europe or region? There are comparing with China and Taiwan, not from the region (Javorac, J. What Are the Effects of Meteorological Factors on Exacerbations of COPD? Atmosphere 2021,12, 442. https://doi.org/10.3390/atmos12040442

Thank you for your question. Yes, there is available literature from Europe and the region as well. We have added the relevant information to the Discussion section. (Line 280.)

The authors cover all aspects of COPD to the Weather Extremities. It would be better to compare to some papers/reviews from the region.

Thank you for your comment. As the Reviewer suggested, we have compared our results to other data reported from the region. (Line 282.)

Reviewer 3 Report

Thank you for the opportunity to review this paper. The authors focused on a current and important topic, however the findings are not entirely new. By the way, the authors of this article came to similar conclusions: Climatic factors influence on emergency department visits on Hong Kong Journal of Emergency Medicine.

The methods are described in detail as regards the collection of meteorological data; it seems rather arbitrary and unclear how to classify the days on the basis of the ED attendance. The statistical methodology with which the results were obtained should be better explained.

The "Results" section should be more concise, the tables contain redundant information, and the key findings are difficult to detect. The discussion is quite complete, however I think this too can be summed up.

In conclusion, considering that 1) the data are 5 years old, 2) that the results are not that innovative and 3) that these can be summarized in a much shorter paper, I believe that the journal could consider to publish this study in the form of a letter to the editor or search letter, if applicable.

Author Response

Dear Reviewer #3,

I would like to thank you for the assessment of our manuscript entitled “How Vulnerable are Patients with COPD to Weather Extremities? – A Pilot Study from Hungary” (healthcare-2015220) and for your insightful suggestions to improve the manuscript. We have addressed each comment and made the requested modifications, where needed.

The authors focused on a current and important topic, however the findings are not entirely new. By the way, the authors of this article came to similar conclusions: Climatic factors influence on emergency department visits on Hong Kong Journal of Emergency Medicine.

Thank you for your comment and for bringing this article to our attention, therefore we have added this recent reference to our manuscript. (Reference No. 11 in Line 46.)

The methods are described in detail as regards the collection of meteorological data; it seems rather arbitrary and unclear how to classify the days on the basis of the ED attendance. The statistical methodology with which the results were obtained should be better explained.

Thank you for your comment. In agreement with the Reviewer, we have elaborated on the methodology in the Methods section, regarding data classification (Line 135.) and we have also clarified the used statistical methodology (Line 149.).

The "Results" section should be more concise, the tables contain redundant information, and the key findings are difficult to detect. The discussion is quite complete, however I think this too can be summed up.

Thank you for your comment. From multidisciplinary considerations, our aim was to represent both the medical and meteorological aspects in our study. This is the reason the text may appear less concise. However, as suggested by the Reviewer, we have identified a sub-section that could be briefer, so changes have been made in section 3.2. (Line 171.). Also, we believe that the tables should be self-contained, which they are.
